# Design and Fabrication of Bulk Micromachined 4H-SiC Piezoresistive Pressure Chips Based on Femtosecond Laser Technology

**DOI:** 10.3390/mi12010056

**Published:** 2021-01-06

**Authors:** Lukang Wang, You Zhao, Yulong Zhao, Yu Yang, Taobo Gong, Le Hao, Wei Ren

**Affiliations:** 1State Key Laboratory for Manufacturing Systems Engineering, Xi’an Jiaotong University, Xi’an 710049, China; wanglukang@stu.xjtu.edu.cn (L.W.); zhaoyulong@xjtu.edu.cn (Y.Z.); yuuyang@stu.xjtu.edu.cn (Y.Y.); gongtaobo@stu.xjtu.edu.cn (T.G.); bsom13@stu.xjtu.edu.cn (L.H.); rw0192@163.com (W.R.); 2Science and Technology on Applied Physical Chemistry Laboratory, Shaanxi Applied Physics and Chemistry Research Institute, Xi’an 710061, China

**Keywords:** 4H-SiC, pressure sensor, femtosecond laser, bulk micromachining

## Abstract

Silicon carbide (SiC) has promising potential for pressure sensing in a high temperature and harsh environment due to its outstanding material properties. In this work, a 4H-SiC piezoresistive pressure chip fabricated based on femtosecond laser technology was proposed. A 1030 nm, 200 fs Yb: KGW laser with laser average powers of 1.5, 3 and 5 W was used to drill blind micro holes for achieving circular sensor diaphragms. An accurate per lap feed of 16.2 μm was obtained under laser average power of 1.5 W. After serialized laser processing, the machining depth error of no more than 2% and the surface roughness as low as 153 nm of the blind hole were measured. The homoepitaxial piezoresistors with a doping concentration of 10^19^ cm^−3^ were connected by a closed-loop Wheatstone bridge after a rapid thermal annealing process, with a specific contact resistivity of 9.7 × 10^−5^ Ω cm^2^. Our research paved the way for the integration of femtosecond laser micromachining and SiC pressure sensor chips manufacturing.

## 1. Introduction

Silicon-based micro-electromechanical system (MEMS) sensors served great markets for pressure sensing in intermediate temperature applications range not exceeding 200 °C [1,2,3]. The thermal ceiling of the silicon piezoresistive pressure sensor is raised to 400 °C due to the adoption of the silicon-on-insulator (SOI) structure, that is, two Si wafers are bonded together by thermally grown silicon oxide (SiO_2_) [4,5]. However, expensive and complicated packaging solutions, such as cooled bushing and heat sink, have to be implemented to mitigate the thermal effect [6]. It is still challenging to achieve long-term pressure measurement on account of the plastic deformation of silicon in extremely severe environments (600 °C), such as automobile combustion chamber, geothermal exploration, gas turbines and aircraft engines.

Silicon carbide (SiC) has long been regarded as an ideal material for high temperature applications thanks to its superior properties over silicon, such as wider bandgap, higher thermal conductivity and higher saturation velocity [7]. There are about 200 different polytypes of SiC, among which only 3C-SiC, 6H-SiC and 4H-SiC could be grown as single crystalline for applications in microelectronics and MEMS devices [8,9]. On the basis of the surface micromachining technology of epitaxial thin film on Si or SOI substrate, 3C-SiC was once the preferred material for high temperature pressure sensors. However, due to the high-density defects caused by lattice mismatch, the nonuniform thickness of epitaxial films and the residual stress, the operating temperature of 3C-SiC pressure sensors does not exceed 400 °C [10,11,12]. With the development of manufacturing techniques of large-scale semiconductor grades of 6H-SiC and 4H-SiC wafers, “all-SiC” pressure sensors based on bulk micromachining technology have been gradually researched. Due to its homogeneity of epitaxy, “all-SiC” pressure sensors avoid the harmful mismatches in the coefficient of thermal expansion [13]. Robert S. Okojie et al. presented a 6H-SiC pressure sensor, which has the ability to work at 600 °C with a 50~65% decline in output signal [14]. Fawzia Masheeb et al. developed a leadless packaged 6H-SiC pressure sensor, which was tested for pressure up to 1000 psi and at temperatures up to 600 °C [15,16]. The bandgap of 4H-SiC is wider than that of 6H-SiC (~3.28 eV), and the electron mobility is close to 3C-SiC (~1000 cm^2^ V^−1^ s^−1^ for electron), which is considered as the most promising SiC polytype in industrial applications under extreme environments [17]. Terunobu Akiyama et al. reported the test results of the 4H-SiC sensor chip directly exposed to a temperature of 650 °C [18]. According to [19], the increase in the uncooled 4H-SiC sensor full-scale output sensitivity at 800 °C implied a higher signal to noise ratio and improved fidelity in actual pressure tests.

However, standard Si bulk micromachining technology has a poor effect for fabricating pressure-sensitive diaphragms in bulk micromachining of “all-SiC” pressure sensors. Wet etching processes used for SiC, including chemical etching and electrochemical etching techniques, have encountered problems such as slow etching rates and severe lateral etching [20]. Plasma etching techniques, including reactive ion etching (RIE), electron cyclotron resonance (RCR) etching and inductive coupled plasma (ICP) etching, are currently the most common approach for patterning and structuring SiC [21,22,23,24]. Some progresses in bulk micromachining SiC using non-standard techniques, such as ultrasonic vibration mill-grinding technique [25], mechanical milling [18] and laser ablation [26,27], have been demonstrated. The emergence of ultrafast lasers (the pulse width is in the range of 10^−15^ to 10^−12^ s, and the energy intensity ranges from 10^13^ to 10^21^ W cm^−2^) overcome the inherent shortcomings of long-pulse lasers, such as rough processing effects and hard-to-eliminate thermal effects, and have gradually become an effective tool for microscale even nanoscale processing of materials such as semiconductors, ceramics and metals [28,29,30,31]. Femtosecond laser has been proved to be an effective method for SiC pressure sensor diaphragm fabrication among them [32,33,34]. However, for brittle materials, the microscopic damage accumulated on the diaphragm caused by the stress wave during femtosecond laser processing is still inevitable [35,36]. Therefore, the laser parameter combination is worth studying to meet the requirement of low-damage and smooth-sensor diaphragms preparation. The integration of laser machining and MEMS technology for SiC diaphragms is rarely reported. In this paper, simulation based on sensor design principles and successful fabrication of the 4H-SiC piezoresistive pressure chip diaphragms with accurate thickness and smooth surface are demonstrated. The Wheatstone bridge configuration located on the front side of the piezoresistive diaphragm of the chip was constructed and tested.

## 2. Materials and Methods

### 2.1. Theory and Modelling

Figure 1 shows a schematic of the bulk 4H-SiC piezoresistive sensor chip with a circular diaphragm (1200 μm in diameter and 80 μm in thickness) used in this work. The flat pressure-sensitive diaphragm structure of the SiC pressure sensor has proven to be simple but effective [14]. The output stress of the circular diaphragm is less than that of the square and rectangular diaphragm, which makes it have a better design space in the high-pressure nonlinear measurement. According to the application requirements of 200% full-scale 5 MPa overload, the sensor has to sacrifice some sensitivity to meet the impact of dynamic fluid. The 2D induced radial and tangential stresses on the surface of the diaphragm at any point are computed with the circular plate small deflection, as expressed below [37]:(1)$$\sigma_{r}=\frac{3P}{8t^{2}}\left[(1+\nu )a^{2}-(3+\nu )r^{2}\right]$$
(2)$$\sigma_{\tau }=\frac{3P}{8t^{2}}\left[(1+\nu )a^{2}-(1+3\nu )r^{2}\right]$$
where σ*_r_* and σ*_τ_* are radial and tangential stress, respectively, P stands for applied pressure load, *t* represents the thickness of the sensor diaphragm, *ν* is the Poisson’s ratio, a is the diaphragm radius, *r* is the distance from any point on the diaphragm to the center of the circle. Within the pressure range, the sensitive diaphragm should operate within the elastic limit. In order to avoid nonlinear relations between stress and pressure, the maximum deflection w*_m_* (at the center of the diaphragm) should be less than 30% of the thickness t of the diaphragm. The breaking stress σ_p_ of SiC is about 492 MPa, and the allowable stress [σ] is 197 MPa if the safety coefficient ns is 2.5. According to the fourth strength theory, when the maximum equivalent stress of the interior and surface of the diaphragm under full range load does not exceed the allowable stress, the design of the sensor diaphragm conforms to the safety principle. Under 200% full-scale pressure overload, the induced stress should not exceed the breaking stress of the material. The above boundary conditions should meet the following criteria:(3)$$w_{m}=\frac{3{Pa}^{4}}{16{Et}^{3}}\left(1-\nu^{2}\right)\leq 0.3t$$
(4)$$\sigma_{eg}=\sqrt{\sigma_{r\;\max}^{2}+\sigma_{\tau }^{2}-\sigma_{r\;\max}\sigma_{\tau }}\leq \left[\sigma \right]$$
where the young’s modulus *E* (~500 GPa) of silicon carbide is isotropic, which is caused by the transversely isotropic property of Hooke’s law applied for the hexagonal lattice [38]. In Equation (4), σ*_r_* _max_ and σ*_τ_* are the maximum radial stress and tangential stress at the edge of the diaphragm, respectively:(5)$$\sigma_{r\;\max}=\frac{3{Pa}^{2}}{4t^{2}}$$
(6)$$\sigma_{\tau }=\frac{3{Pa}^{2}}{4t^{2}}\nu$$

The isotropy of the piezoresistive effect of 4H-SiC in each crystal direction of the (0001) plane is attributed to its wurtzite lattice in the 6mm point group [39]. Nevertheless, the absolute value of the longitudinal piezoresistive coefficient π_11_ is larger than the transverse piezoresistive coefficient π_12_ [40]. The piezoresistors arranged tangentially on the circular diaphragm, with current flowing parallel to it, will experience a longitudinal stress induced by the tangential strain component. The transverse stress caused by the radial strain component (strain perpendicular to the piezoresistor) will also act on the piezoresistors and introduce a negative piezoresistive coefficient. In contrast, the piezoresistors arranged radially on the circular diaphragm will be dominated by the longitudinal stress generated by the radial strain component. At the same time, the tangential stress combined with negative piezoresistive coefficient introduces a faint transverse effect. Therefore, the piezoresistors should be located in the regions of the maximum stress and longitudinal piezoresistance effect in order to achieve optimum piezoresistive properties. Figure 2a,b shows the von Mises stress and the displacement variation along the radial direction on the diaphragm under different pressure loads, which are simulated in COMSOL Multiphysics software. The piezoresistors will be placed in the region of stress concentration at the edge and center of the diaphragm. In order to utilize the maximum induced stress region and obtain a consistent relative change in resistance, the piezoresistors at the edges of diaphragm are folded, while that at the center of the diaphragm is unfolded. The output voltage of the Wheatstone bridge changes with the distance of the folded piezoresistors from the center of the circular diaphragm, and the piezoresistors have different sizes but the same resistance value, as shown in Figure 2c. When the four sizes of piezoresistors are in the optimum position, the sensitivity of the sensor output is similar, as shown in Figure 2d. Considering the error of the photolithography and etching process, the piezoresistor size is selected as 200 × 20 μm. Under the coupling effect of solid mechanics and electric current-single layer shell field, the diagram of the von Mises stress and the electric potential is shown in Figure 3.

### 2.2. Laser Micromachining for Diaphragms

In MEMS applications, the pointing stability of the laser output beam is a very vital issue for micromachining features. The laser pulse generated by the ultrafast laser oscillator is utilized to meet the rigorous precision requirement. As shown in Figure 4, a laser micro-processing system equipped with a femtosecond Yb: KGW laser (Light Conversion, Pharos, Lithuania) is available for structuring bulk 4H-SiC sensor diaphragms. On-demand pulse operation is possible due to the existence of the laser pulse picker. The specification of the laser output beam is shown in Table 1. The laser beam passing through the wave plate is collimated and circular polarized. It is then modulated by a beam modulator, which operates as a shutter to turn on and off the laser output when required. The quality of the pulsed laser beam is improved by a diaphragm, which serves to cut the edge of the Gaussian beam. After passing through two mirrors and a polarizer, the laser beam is then scanned on the X and Y axes by a two-axis galvanometer scanner in a mode of direct writing a series of crossing lines passes. The 4H-SiC sample, which is placed at a distance equal to the working distance of the focusing lens from the output of the lens, is fixed on the mechanical translational stage. During laser scanning, the mechanical translational stage steps in the opposite direction of the *Z* axis. Therefore, the laser focus is always on the surface of the 4H-SiC sample, and the size of the laser spot remains the same. A scanning electron microscope (SEM, SU8010, Hitachi, Japan) is employed to examine the oblique view and top view of the samples. The depths of the micro blind holes were measured by a laser scanning confocal microscopy (LSCM, OLS4000, OLYMPUS, Tokyo, Japan) without destroying the samples. The surface roughness *Ra* of the samples was characterized by an atomic force microscope (AFM, Innova-IRIS, Bruker, Germany).

### 2.3. Wheatstone Bridge Configuration Fabrication Process

The manufacturing process of the 4H-SiC piezoresistive chips is shown in Figure 5. A 2 μm thick n-type epitaxial layer with a doping concentration of 10^19^ cm^−3^ and a 5 μm thick p-type epitaxial layer with a doping concentration of 10^18^ cm^−3^ were grown on the 4° off-axis basal (0001) plane of a 4 inch and 350 μm thickness 4H-SiC wafer, which was produced by Epi World International Co., Ltd., Xiamen, Fujian, China. After laser processing, the SiC sample was immersed in HCl:H_2_O_2_:H_2_O (5:3:3) for 5 min and then in HF:HNO_3_ (1:1) for 5 min to remove the oxide particles and other debris accumulated on the surfaces. Due to the tremendous optical transparency of the single crystal 4H-SiC wafer, the photolithography alignment on the front of the SiC substrate for Wheatstone bridge fabricating is easy to realize. A nickel (Ni) evaporation and lift-off process was performed on the top n-type 4H-SiC epilayer with photoresist. After lift-off, the Ni layer with a thickness of 200 nm served as an etching mask to define the pattern of the piezoresistors. The piezoresistors were fabricated by ICP etching using Oxford ICP-180 system, and its process parameters are shown in Table 2. An etching depth of 2.0 μm was measured by an AMBIOS XP-2 profiler (AMBIOS Technology Corporation, Milpitas, CA, USA), which ensured that the unmasked n-type functioning layer was thoroughly etched, and the robust p-n junction isolation was formed. The remaining Ni was stripped from the raised resistances in a mixture of HNO_3_:HCl:H_2_O (1:5:3). As a passivation layer, silicon dioxide (SiO_2_) was deposited with a thickness of 200 nm by plasma enhanced chemical vapor deposition process (PECVD, Minilock-Orion, Trion Technology, Clearwater, FL, USA). Contact holes were opened by photolithography and wet etching in the SiO_2_ to expose the ends of the piezoresistors that acted as lead terminals. High quality Ohmic contact is essential for the transmission of electrical signals between semiconductor and metal circuits. Multilayers of Ni (100 nm)/Ti (50 nm)/Pt (200 nm) were sputtered on the 4H-SiC as metal contact layer and lead connection layer. Ni in the multilayer metal reacts with SiC to form NiSi_2_, which lowers the barrier at the interface to form Ohmic contact [41]. Ti reacts with surplus C at the interface to form Ti_3_SiC_2_ to improve the stability of Ohmic contact [42]. The purpose of Pt is to protect the underlying metal from oxidation during annealing. This was followed by a rapid thermal annealing process using UNITEMP RTP-100 (UniTemp GmbH, Pfaffenhofen, Germany) in nitrogen at 500 °C for 5 min and 1000 °C for 3 min to realize Ohmic contact.

## 3. Results and Discussion

Dimensional accuracy, surface smoothness and throughput of sensor diaphragms are three aspects worthy of attention in the laser manufacturing process. Stability is essential to control and obtain the repeatability of the ablative feature dimensions. Thanks to the diode pumped solid state (DPSS) laser technology, the system is stable in terms of laser power and pulse-to-pulse energy. The thickness of the sensor diaphragm depends on the depth of the micro blind holes. With other parameters unchanged (as shown in Table 1), the depths of micro blind holes are affected by lap and laser scanning speed when the laser average power is fixed, as shown in Figure 6a. The lap is defined as the repetitions of scanning a completed circular diaphragm region with a laser beam. Under different laser average powers and laps, the machining depth decreases with the increase in laser scanning speed, as shown in Figure 6b–d. The average material removal depth per lap is 16.2, 33.1 and 79.2 μm/lap when the laser average power is 1.5, 3 and 5 W, respectively. Therefore, a more accurate per lap feed can be obtained under laser average power of 1.5 W. Under this energy, further parameter optimization is carried out by adjusting the laser scanning speed to achieve the required depth of micro blind hole, as shown in the blue line in Figure 6d. As shown in Figure 7, the depth error of the 8 micro blind holes is not more than 2%. They were all processed with the same set of optimized experimental parameters: laser average power of 1.5 W, lap of 15 and laser scanning speed of 125 mm/s. The micromorphology of the blind hole is shown in Figure 8a,b. The best surface roughness *Ra* could as low as 153 nm. Although femtosecond laser processing is well known for its characteristics of small-heat affected zone, it still causes heat effect due to the heat accumulation, especially when the repetition rate is higher than 10 kHz [43,44]. In order to verify that the accumulated heat does not affect the crystal structure of the epitaxial 4H-SiC layer for pressure–electrical signal conversion on the front side of the diaphragm, additional X-ray measurements were performed using an automated Bragg–Brentano diffractometer (D8 ADVANCE A25, Bruker, Germany) in a step-scanning mode with the scanning step size of 0.02°. A Cu tube with 1.5418 Å was applied with the tube working conditions 40 kV and 40 mA. As shown in Figure 8e, the raw sample without laser processing is used as the control group, and its pattern has a sharp diffraction peak at 35.80°. In comparison, the epitaxial layer on the front side of the silicon carbide sample after laser processing has a similar XRD pattern, which proves that the accumulated heat during laser processing of blind holes does not affect the crystal structure of the other side. The XRD pattern of the laser irradiated region shows that the diffraction peak intensity of 4H-SiC (0004) crystal plane is weakened, and the diffraction peak of silicon appears. This is caused by the thermal reaction between silicon carbide and oxygen under the action of laser energy. The possible chemical reactions can be described as: SiC + O_2_ → Si + CO_2_; SiC + O_2_ → SiO + C; SiC → Si + C. Although, the laser machining is inherently a serial process, where the processing time is strongly dependent on the pattern loads and scanning laps. The femtosecond laser power is high enough to meet the industrial throughput under real conditions. The average processing time of each micro blind hole is about 1 min using our optimized experimental parameters. The net throughput could be ultimately comparable with the highly parallel dry-etching (ICP, RIE, ECR etching and so on) and wet-etching process where, however, a reliable mask and anisotropy are tricky issues.

A Keysight B1500A semiconductor analyzer (Keysight Technologies Inc., Santa Rosa, CA, USA) was used to measure the relationship between the current (I) and voltage (V) of the four 4H-SiC resistances in the open-loop Wheatstone bridge of the sensor. The I-V curves, as shown in Figure 9a, illustrate that good Ohmic contact between the Ni layer and the n-type SiC epitaxial layer was realized after the rapid thermal annealing. Furthermore, the measurement of the leakage current from n-type SiC epitaxial layer to the SiC substrate indicates that the strong p-n junction provided excellent isolation with a slight current leakage of less than 61 nA at an applied DC voltage of up to 5 V, as Figure 9b shows. Figure 9c shows the surface morphology of the compact annealed metal near the contact window. Four piezoresistors in the Wheatstone bridge have resistances of 1100, 1051, 1093 and 1086 Ω, respectively, which are close to the designed resistance of 1000 Ω. The error of the resistance values mainly comes from the contact resistances. The specific contact resistance (SCR) measured based on the linear transmission line model (LTLM) [45] is 9.7 × 10^−5^ Ω cm^2^, as shown in Figure 9d. Compared with the SCRs in Table 3, such a low value (in the range of 10^−4^ to 10^−5^ Ω cm^2^) indicates that a good metal–semiconductor connection has been achieved. It also proves that the electrical characteristic of the bridge resistance on the front of the sensor chip is not affected by the residual effects of laser processing. The integrated manufacturing sensor chips array, not yet diced, as shown in Figure 9e. Further dynamic pressure testing will be carried out in future work.

## 4. Conclusions

In this paper, a 4H-SiC piezoresistive pressure chip fabricated based on femtosecond laser machining technology was reported. The integration of the MEMS process on the front side of the SiC pressure-sensitive chip and the laser processing technology on the backside is introduced. The fourth strength theory is adopted, considering the application requirements of 200% full-scale 5 MPa overload. Finite element analysis was performed using COMSOL Multiphysics software to determine the best configuration of four piezoresistors in the Wheatstone bridge. The best laser parameter combination was obtained when the laser average power was 1.5 W, which achieved the smallest depth error and surface roughness of the sensor diaphragms. Good Ohmic contact was obtained with a specific contact resistivity of 9.7 × 10^−5^ Ω cm^2^ after the rapid thermal annealing process. In future research, the pressure calibration and high temperature performance test of the 4H-SiC sensor chips will be further studied.

## Figures and Tables

**Figure 1 micromachines-12-00056-f001:**
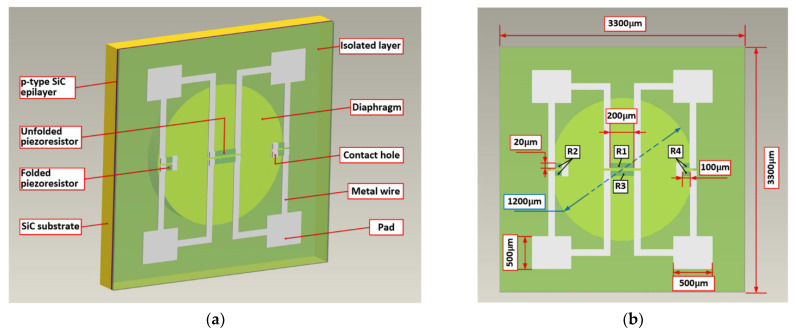
(**a**) Schematic diagram and (**b**) dimension diagram of the 4H-SiC piezoresistive sensor chip.

**Figure 2 micromachines-12-00056-f002:**
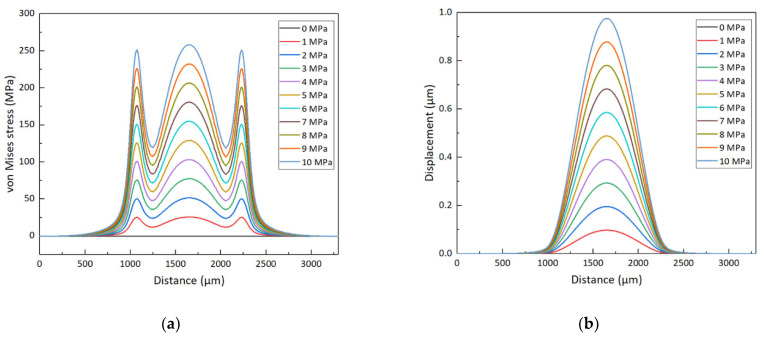
(**a**) The von Mises stress and (**b**) the displacement along the radial direction of the sensor chip diaphragm under different pressure loads. (**c**) The outputs of the four Wheatstone bridges with piezoresistor of different sizes vary with the distance between the folded piezoresistor and the center of the circular diaphragm. (**d**) The outputs of the Wheatstone bridge under 200% full-scale overload pressure.

**Figure 3 micromachines-12-00056-f003:**
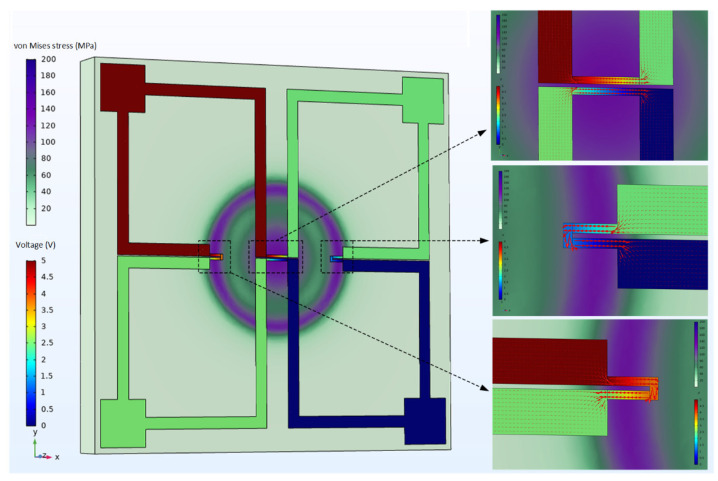
Coupled-field diagram of solid mechanics and electric current (single layer shell). The red arrow is the surface current density of the piezoresistors in the three small graphs on the right.

**Figure 4 micromachines-12-00056-f004:**
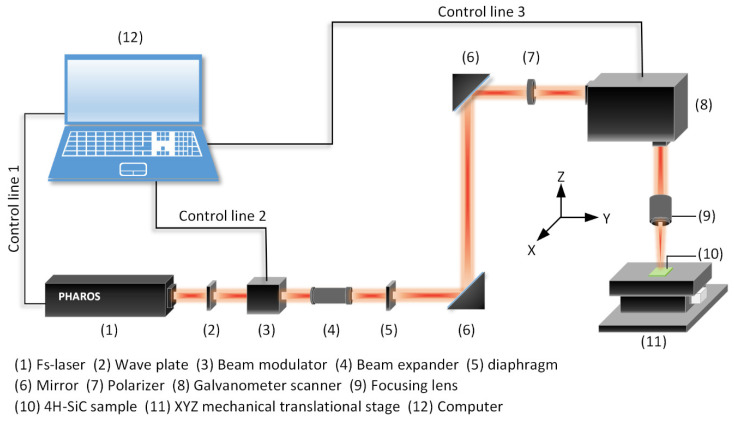
Experimental setup for femtosecond laser micromachining of the 4H-SiC sample.

**Figure 5 micromachines-12-00056-f005:**
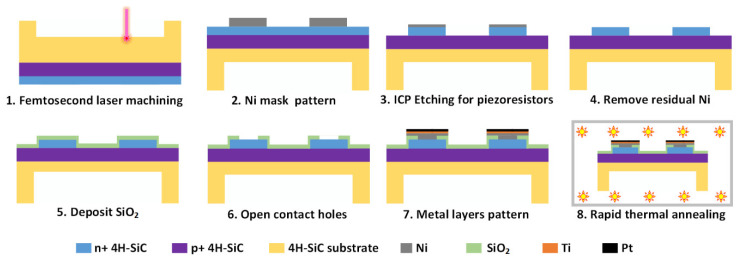
Schematic process of manufacturing 4H-SiC piezoresistive pressure sensor chip.

**Figure 6 micromachines-12-00056-f006:**
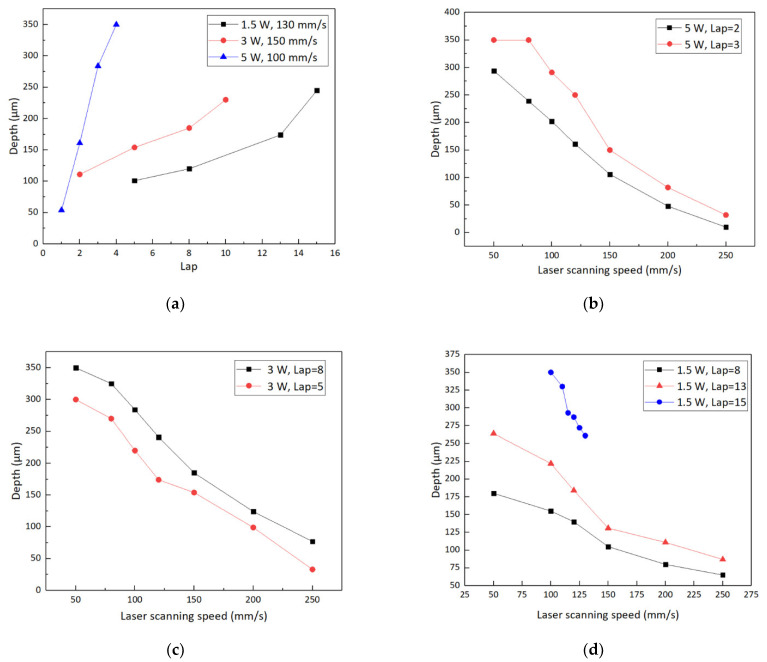
(**a**) Depths of micro blind holes as a function of lap for laser average power of 1.5, 3 and 5 W. Depths of micro blind holes depending on laser scanning speed under different laser average powers and laps: (**b**) laser average power of 5 W, laps of 2 and 3; (**c**) laser average power of 3 W, laps of 5 and 8; (**d**) laser average power of 1.5 W, laps of 8, 13 and 15.

**Figure 7 micromachines-12-00056-f007:**
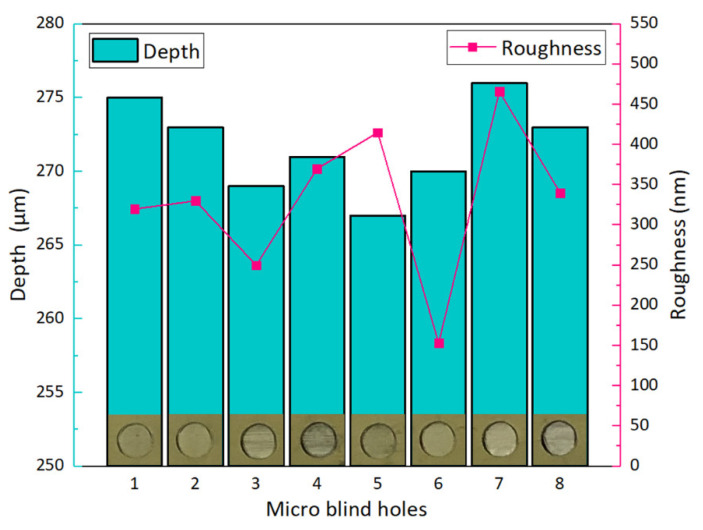
Depth and roughness of micro blind holes, which were all processed with the same set of parameters: laser average power of 1.5 W, lap of 15 and laser scanning speed of 125 mm/s.

**Figure 8 micromachines-12-00056-f008:**
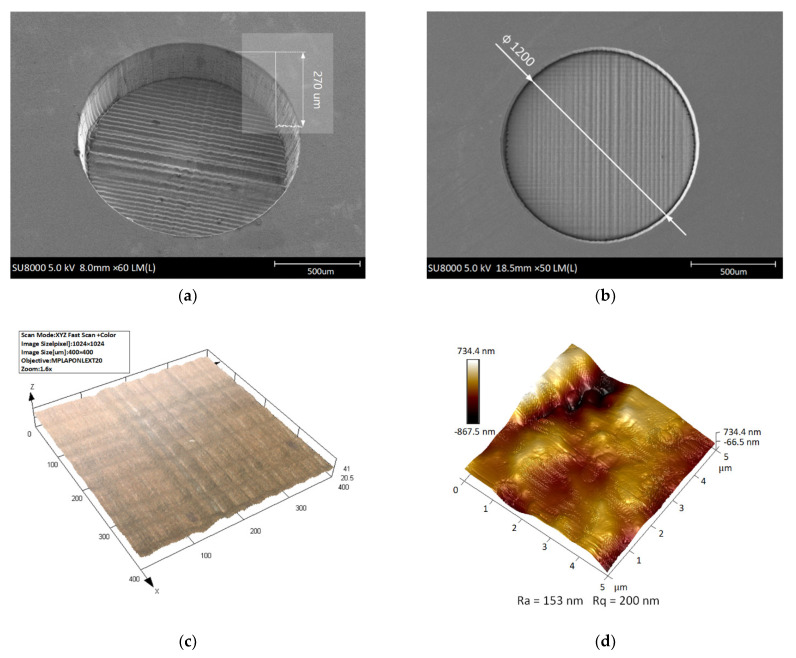
SEM images of micro blind hole in oblique view (**a**) and top view (**b**). (**c**) Laser scanning confocal microscopy (LSCM) image of the bottom of the micro blind hole. (**d**) Atomic force microscope (AFM) 3D image of the surface of laser-irradiated region. (**e**) XRD diffractograms of different 4H-SiC samples.

**Figure 9 micromachines-12-00056-f009:**
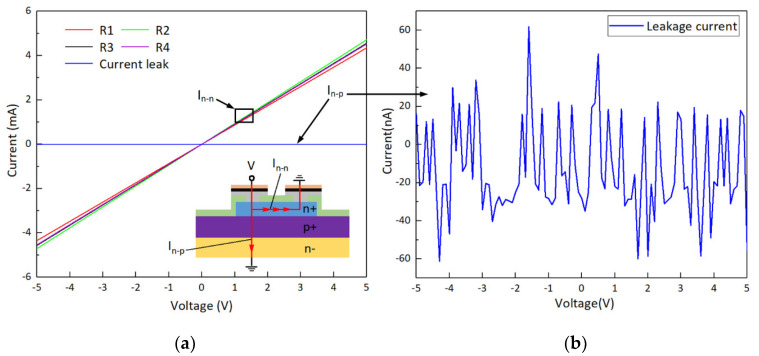
(**a**) Current–voltage characteristics of the piezoresistors in a sensor open loop Wheatstone bridge. Inset: Schematic diagram of the measurement of the current flowing in the n-type epitaxial 4H-SiC layer (*I*_n-n_) and through the p-n junction (*I*_n-p_). (**b**) Slight current leakage measurement. (**c**) SEM image of the metal surface morphology after annealing. (**d**) Linear transmission line model (LTLM) patterns for specific contact resistance (SCR) measurement. (**e**) Photograph of the 4H-SiC pressure sensor chips array.

**Table 1 micromachines-12-00056-t001:** Process parameters of the femtosecond laser.

Process Parameters	Values
Center wavelength	1030 nm
Pulse duration	200 fs
Repetition rate	100 kHz
Jumping speed of galvanometer	8000 mm/s
Scanning line interval	2 μm
Laser spot diameter	30 μm
Beam quality	TEM_00_ (M_2_ < 1.2)
Beam pointing stability	<20 μrad/°C
Power stability	<0.5% rms over 100 h
Output pulse-to-pulse stability	<0.5% rms over 24 h

**Table 2 micromachines-12-00056-t002:** Process parameters inductive coupled plasma (ICP) etching.

Process Parameters	Values
Gas conditions	O_2_:CF_4_ (10 sccm:35 sccm)
ICP power	800 W
Radio frequency (RF) power	100 W
Chamber pressure	10 mTorr
Direct current (DC) bias	−120 V
Average etching rate	~200 nm/min

**Table 3 micromachines-12-00056-t003:** Summary of results reported on specific contact resistivity of 4H-SiC Ohmic contact.

Contact Metal	Doping Concentration (cm^−3^)	SCR (Ω cm^2^)	Reference
Ni	1 × 10^20^	1 × 10^−4^	[46]
Ni/Ti/Al	3.0~9.0 × 10^18^	6.6 × 10^−5^	[47]
Ni/Al/Ti/Ni	3.0~4.0 × 10^19^	1.5 × 10^−5^	[48]
Al/Ti	1.0~4.0 × 10^19^	2.7 × 10^−4^	[49]
Ni/Ti/Ni/TaSi_2_/Pt	1.3 × 10^19^	3.7 × 10^−4^	[50]

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
