# Peer review of "Design and Fabrication of Bulk Micromachined 4H-SiC Piezoresistive Pressure Chips Based on Femtosecond Laser Technology"

_micromachines, 2021, doi:10.3390/mi12010056_

Round 1
Reviewer 1 Report
The manuscript “Design and fabrication of bulk micromachined 4H-SiC piezoresistive pressure chips based on femtosecond laser technology” by L.Wang et al. describes in all details the fabrication a 4H-SiC piezoresistive pressure chip by means of a femtosecond laser technology. Firstly, simulation based on sensor design principle are performed and, then, the different steps of the fabrication process of the 4H-SiC piezoresistive pressure chip diaphragms are demonstrated.
The manuscript results clear and detailed, therefore I suggest to accept it for publication in the present form.
Minor revisions required:
In Fig.6b probably the legend is wrong (red and black markers appear to be reversed);
Fig.8d: please explain the meaning of Ra and Rq;
Fig.9 a,b: probably Fig.9 b concerns In-p , not In-n
Reviewer 2 Report
The authors designed and fabricated piezoresistive pressure chips by drilling 4H-SiC using femtosecond laser. The numerical simulation was well organized and pressure chip was well fabricated. However, the novelty of this paper was not clearly stated. The authors stated that “the integration of laser machining and MEMS technology for SiC diaphragms is rarely reported”. If this integration is the novelty of this paper, then the authors should more carefully investigate the femtosecond laser processing conditions or describe how the parameters are optimized. The other parts are well-written. Therefore, I would like to recommend it for a publication after the following questions and comments are addressed by the authors.
1. Line 65, Page 2. Although femtosecond laser is a good tool for microprocessing materials, damage can be actually generated when brittle materials are machined [Appl. Phys. A 126, 861 (2020)], [Appl. Phys. A 124, 181 (2018)]. The authors should comment on the difficulty in using femtosecond laser as well in the manuscript. This may be related to the “smoothness” written in Line 197, Page 7.
2. Line 161, Page 5. Which surface roughness parameter (i.e. Ra, Rz, or Sa) was used? Please specify it.
3. Line 198, Page 7. The authors state “three aspects worthy of attention”. What about the crystalline nature? Although femtosecond laser processing is well-known for its characteristics of small-heat affected zone, it still causes large heat effect especially when the repetition rate is higher than 10 kHz, because of the heat accumulation [Opt. Express 13(12), 4708 (2005)], [Opt. Express 28(10), 15240 (2020)]. The heat effect may impair the crystalline nature, but doesn’t it affect the property of the sensor?
4. Line 198, Page 7. Although the authors stated the three aspects (accuracy, smoothness, and throughput) are important, it seems the authors optimized the parameters only from the point of view of accuracy. The throughput can be increased if the laser power is much higher. There are many other parameter sets even if the power is kept the same such as by increasing the pulse energy and decreasing the repetition rate. If using femtosecond laser is the novelty of this paper, then the authors should carefully describe this optimization part. If the performance of the developed sensor is the novelty of this paper, then the authors may not need to care about this optimization part seriously. (Please see the comment #7).
5. Fig. 6(b) may be wrong? Why was the depth smaller when Lap = 3?
6. Line 228, Page 8. The caption “laps of 8, 13 and 8” should be “laps of 8, 13 and 15”.
7. Line 249, Page 9. How good is the value of 9.7×10-5 Ω cm2 compared to the reference?
Round 2
Reviewer 2 Report
The authors have carefully modified their original manuscript. I now would like to recommend it for a publication.